# Study of the Reliability of Field Test Methods for Physical Fitness in Children Aged 2–3 Years

**DOI:** 10.3390/ijerph19127522

**Published:** 2022-06-20

**Authors:** Dandan Ke, Duona Wang, Hui Huang, Xiangying Hu, Jun Sasaki, Hezhong Liu, Xiaofei Wang, Dajiang Lu, Jian Wang, Gengsheng He

**Affiliations:** 1School of Public Health, Fudan University, Shanghai 200032, China; kedandan@fudan.edu.cn; 2Graduate School of Health and Sports Science, Juntendo University, Chiba 2701695, Japan; 3International College of Football, Tongji University, Shanghai 200092, China; 2032079@tongji.edu.cn; 4Shanghai Jingan Center for Women and Children’s Health, Shanghai 200062, China; huanghui@jinganfuyou.onaliyun.com (H.H.); huxiangying@jinganfuyou.onaliyun.com (X.H.); 5Kao (China) Research & Development Center Co., Ltd., Shanghai 200241, China; sasaki.jun@kao.com (J.S.); liu.hezhong@kao.sh.cn (H.L.); 6School of Kinesiology, Shanghai University of Sport, Shanghai 200438, China; 1811517004@sus.edu.cn (X.W.); ludajiang@sus.edu.cn (D.L.)

**Keywords:** physical fitness, physical activity, growth and development, fundamental motor skill, preschool children

## Abstract

Physical fitness measures overall physical health. It is the ability of the body to work effectively and stay healthy during leisure and emergencies. Given the progressive integration of 2–3-year-olds into preschool, physical fitness testing of these children has become increasingly important. We aimed to develop and test the reliability of an appropriate field test method for physical fitness in 2–3-year-olds children. One hundred and three children (44 boys and 59 girls) volunteered for this study. Their height and weight were tested, and the same tester conducted the test twice for handgrip strength, 3 m balance walking, stair climbing, 5 m run, and kicking a ball at one-minute intervals. Pearson correlation coefficient and intraclass correlation coefficient (ICC) were used for reliability testing. The reliability of this field test method for physical fitness was high in the repetitive tests of Chinese 2–3-year-olds for the four items of handgrip strength, 3 m balance walking, stair climbing and 5 m run, and the reliability was moderate for the kicking the ball item. This study indicates that these field-based physical fitness test methods have good reliability and are simple, feasible, safe, and easy to be accepted and understood by 2–3-year-old children; thus, it may be used as a reference for professionals in China and abroad.

## 1. Introduction

Physical fitness refers to the ability of the body systems to work effectively in harmony to enjoy leisure time, stay healthy, and cope with emergencies. It comprises of two components of physical fitness: health-related physical fitness and skill-related physical fitness [1,2]. The basic requirements for being physically fit are the absence of illnesses, a well-formed and well-developed body, good physiological development, good physiological functioning of the body systems and the ability to be highly physically active. Numerous studies have found that physical fitness has a strong relationship with psychological, cognitive, and academic performance [3,4,5], in addition to being associated with a decreased prevalence of various non-communicable diseases (obesity, type 2 diabetes, etc.) [6,7,8,9,10]. Meanwhile, in recent years, individuals have become less physically active due to the changing nature of many jobs, more convenient transportation, and increased urbanization, which has led to an overall decline in physical health, especially among children and adolescents [11,12].

The physical health of a country’s population is an important component of the overall national power of the country. Preschool children are at a stage of rapid development, which will have a significant impact on their lifelong well-being, physical development, and psychological health [13,14]. Being in good physical condition is not only a prerequisite for the healthy growth of children, but also a critical factor for talent selection and monitoring of development in sports [15,16,17]. The preschool period is a critical period in the shaping of a person’s physical fitness level and has a significant impact on the level of physical health in adulthood; hence, focusing on the health of young children is beneficial in improving the overall health of a country’s population. As such, a validated set of physical health tests is particularly important for the day-to-day management of young children’s health. Conducting laboratory tests with large samples is generally not feasible due to their high cost in population-based fitness studies. Conducting field tests is the alternative that is available at a relatively low cost [18].

Currently, a large number of field test methods for children and adolescents in China and abroad (e.g., ALPHA [19], EUROFIT [20], FITNESSGRAM [21], PreFit [22,23] and the fitness test battery by Latorre [24,25]), as well as the Chinese National Physical Fitness Measurement (CNPFM-Pre) and the National Physical Fitness Test Program released by the Chinese State General Administration of Sports in 2000 [26,27,28] have been proven to be valid and are widely used. At age 2, children are beginning to develop various basic motor skills [13,14] and have acquired a certain level of learning ability. Childcare is being progressively integrated into preschool education in many countries. Although there are internationally recognized assessment scales such as ASQ [29,30,31] that are used to assess the motor development of children under 3 years of age, they require one-on-one assessment by professionals such as physicians, which is a cumbersome and time-consuming procedure that cannot be implemented in a preschool setting. Therefore, the development of a field test method to evaluate the physical fitness of 2–3-year-olds is particularly important during this time of rapid development in the childcare industry.

Based on this, our study developed a set of physical fitness methods for 2–3-year-old children according to their growth and development characteristics. This study set out to test the hypothesis that the field-based physical fitness test methods may have good reliability in 2–3-year-old children, to provide a reference testing protocol for research and practice in this field.

## 2. Materials and Methods

### 2.1. Study Design and Subjects

G-power 3.1 was used for prior analysis and the total sample size was estimated to be 82. Based on the class-based block group entry and informed parental consent, a total of 103 children (44 boys and 59 girls) aged 2–3 years from Chinese nursery schools were enrolled in this study. Written informed consent forms were obtained by the parents or legal guardians prior to the start of the tests to inform them of the study content and potential risks. The study was approved by the Shanghai Nutritional Medicine Ethics Committee (Approval No. 2021–08) on 24 May 2021.

### 2.2. Materials and Tests

According to the generally accepted classification of physical fitness components [1,2], the test items were designed to cover both health-related and skill-related physical fitness domains. As shown in the proposal of field-based physical fitness test for 2–3-year-old (Figure 1), health-related physical fitness items include body composition (body mass index, BMI) and handgrip strength, and skill-related physical fitness items include 3 m balance walking (s), stair climbing (s), 5 m run (s), and kicking the ball (m).

#### 2.2.1. Body Composition

Both height and weight were measured while wearing no shoes with lightweight clothing on. The height of a person reflects the level of longitudinal growth of the human skeleton. The test was performed with the participant standing barefoot and in an upright position on the base plate of the height meter (an upright torso and naturally drooping upper limbs, heels together and toes about 60° apart), with the heels, sacrum and both scapulae in contact with the post of the height meter, with head upright, both eyes looking straight ahead, and the upper edge of the ear screen level with the lowest point of the lower edge of the eye socket. The recorded measurements were in centimeters rounded to one decimal place. The body weight of a person reflects the degree of development and nutritional status of the body. The test was performed with the participant standing naturally at the center of the scale, and the data were read after he/she stood firmly. The recorded data were expressed in kilograms rounded to one decimal place. Body mass index BMI (kg/m^2^) was calculated as the weight divided by the square of the height.

#### 2.2.2. Handgrip Strength

The handgrip strength test primarily reflects upper limbs strength and is tested by the handgrip strength meter device (TKK model 5001, Grip-A, Takei, Tokyo, Japan). During the test, the device was positioned with the pointer pointing outward and adjusted according to the size of the palm so that the second joint of the index finger was at a right angle before taking the measurement. The body was kept upright with the feet separated naturally; the grip meter was not touching the body or clothes as far as possible, the handgrip strength meter should not swing back and forth when measurements are being taken, and the body should be kept as still as possible during measurement. Measurements were taken in the order of right first and then left, and each hand was measured twice, taking the best result which was to be recorded in kilograms, and each child could have an attempt within the 2 min time limit, otherwise a missing score was recorded. Test result below 0.5 kg was considered invalid and would need to be re-measured. If the test time is longer than 2 min, a missing score was recorded. The point of teaching 2–3-year-old children is to let them understand how to force their hands to grip the device. The nursery health teacher will let them feel the sensation by gripping their arms and tell them that the tighter the grip, the farther the needle on the device will run and the better it will be.

#### 2.2.3. 3-m Balance Walking

The 3 m balance walking mainly reflects the balance ability of young children, which is tested through a 3.0 m-long and 0.2 m-wide walkway and a stopwatch. During the test, the participant would stand with both feet together behind the starting line, and he/she would walk in a straight line after hearing the “start” command. The tester would start the stopwatch according to the participants’ movement and stop it when they reached the ending line. The recorded data were in seconds with two decimal places. Before the test began, the children were told to walk through the 3 m like a one-way bridge, and to protect themselves from falling into the river during the process. If there was a mistake in the process, the child could try to walk again within one minute.

#### 2.2.4. Stair Climbing

The stair climbing test mainly reflects the coordination ability. The test was conducted using a five-step staircase with specifications of 68 × 27 × 12 cm per step. Before the test, the participant would stand facing the stairs, and after hearing the “start” command, he/she would climb upwards with both feet alternately without holding the handrail, starting from the first step until both feet passed the fifth step. The time taken to climb the steps was recorded, tested twice, taking the lower value, which was to be recorded in seconds up to two decimal places. If there was a mistake in the process, it was possible to test again within one minute.

#### 2.2.5. 5-Meter Run

The 5 m run test primarily reflects the ability of speed. This test is conducted on an outdoor playground, which requires a double track of one meter width and with 5 m separation between the start and finish line, in addition to a 1 m buffer outside the finish line. We put markers in the buffer zone and raced in pairs to motivate children to rush to the finish line as fast as possible. During the test, the participants, in groups of two, would stand in front of the starting line in a standing starting position. When the “run” command was heard, they would start running to the end line with all their might. The tester would start the timer according to the participants’ movement. When the participant’s chest touched the vertical surface of the starting line, the tester would stop the timer. The recorded data were in seconds with two decimal places.

#### 2.2.6. Kicking the Ball

The kicking the ball test reflects the coordination and power of the lower limbs. The test was conducted in a 15 × 5 m field, using a soccer ball with a circumference of 62–63.5 cm and a weight of 320–340 g. During the test, the soccer ball was placed at the middle of the starting line, and the participant stood next to the soccer ball, and a tester standing on the far side signaled the child to kick the ball in the right direction. The participant kicked the ball after hearing the “start” command; the vertical distance from the starting line within the field where the ball stopped or went out of bounds was measured and tested twice, taking the larger value, which was to be recorded in meters with two decimal places. The maximum distance to be taken was 15.0 m. If the participant failed to hit the soccer ball or made other mistakes, he/she could have multiple attempts within the 1 min time limit.

### 2.3. Procedure

The physical fitness tests were conducted after permission was obtained from the nursery school and informed consent was obtained from the parents. Prior to the official test, materials on the test procedure and physical fitness test methods were given to the nursery health teachers, and the test site and related instruments were set up and prepared one day in advance, as well as on-site training for the nursery health teachers by the research team members.

To ensure that each test met the requirements of being tested by the same person, two researchers and one health teacher worked as a team, with one researcher specializing in the measurement of scores, the other researcher arranging the test procedure, and the nursery health teacher assisting in teaching the children the specific test methods. The test process is shown in Figure 2; each child was tested in the order of indoor items (➀ height, weight, ➁ handgrip strength, ➂ 3 m balance walking, and ➃ stair climbing), and outdoor items (➄ 5 m run and ➅ kicking the ball), and the whole test process took about 20 min. The same tester conducted the test twice for each item at one-minute intervals, and the child was encouraged throughout the tests.

### 2.4. Statistical Analysis

The results are expressed as means and standard deviations (SD), and the reliability of the five tested items was tested using Pearson correlation coefficients with intraclass correlation efficient (ICC) and IBM SPSS 26.0 for Windows (IBM Corp., Armonk, NY, USA) with 95% confidence intervals (95% [CI]). ICC values less than 0.5, between 0.5 and 0.75, between 0.75 and 0.9, and greater than 0.90 were considered to have poor, moderate, good, and excellent relative reliability, respectively. Sample size calculations were conducted by G-Power (version 3.1.9.7, University of Kiel, Kiel, Germany), using *t* test (correlation: point-biserial model) with an alpha value of 0.05 and power of 0.8. Comparisons between age and gender were made using an independent samples *t*-test, with significance defined as *p* < 0.05.

## 3. Results

### 3.1. Characteristics of the Participants

The results of the children’s tests were compared according to sex and age, as shown in Table 1. In terms of sex, there were 44 boys and 59 girls who were enrolled in the test, and the results of each test showed no significant difference between sex (*p* > 0.05). In terms of age, 3-year-olds significantly outperformed 2-year-olds in height, weight, 3 m balance walking, 5 m run, stair climbing, and kicking the ball (*p* < 0.01); however, there was no significant difference in handgrip strength.

### 3.2. Reliability of Physical Fitness Test Items

Table 2 shows the repeated test results of the five physical fitness test items. The ICC values were 0.857 (*p* < 0.001) for handgrip strength, 0.821 (*p* < 0.001) for 3 m balance walking, 0.844 (*p* < 0.001) for 5 m run, and 0.791 (*p* < 0.001) for stair climbing, all of which had good repeatability. The ICC value for kicking the ball was 0.672 (*p* < 0.001), which was of average reproducibility.

The correlation between the results of the two tests for the 103 participants is shown in Table 2 and Figure 3, with the horizontal axis showing the results of the tests on the young children and the vertical axis showing the results of the second test. Similar results can be seen in the graphs, with strong correlations shown for handgrip strength, 3 m balance walking, 5 m run, stair climbing, and kicking the ball, where Pearson correlation coefficients were 0.859, 0.821, 0.793, 0.844, and 0.670, respectively (*p* < 0.01).

### 3.3. Correlations between the Physical Fitness Test Items

The better result of each item in both tests was taken for correlation analysis between items, and the results are shown in Table 3. There was a weak correlation (*p* < 0.01) shown for height with handgrip strength, 3 m balance walking, 5 m run and stair climbing, whereas there was a very weak or no correlation shown for weight and BMI with other physical qualities. There was a moderate correlation of handgrip strength, 5 m run and stair climbing with each other (*p* < 0.01). There was also a moderate correlation between 3 m balance walking, 5 m run and stair climbing (*p* < 0.01).

## 4. Discussion

In our study, the field test method for the physical fitness of 2–3-year-olds contained five items: handgrip strength, 3 m balance walking, stair climbing, 5 m run, and kicking a ball. Of which, the reliability was high in the repetitive tests of Chinese 2–3-year-olds for the four items of handgrip strength, 3 m balance walking, stair climbing and 5 m run (as shown in Table 2). Although the reliability was lower for the kicking the ball item, it was still acceptable. This test protocol incorporates the conventional test items of height, weight, and BMI to provide a more comprehensive quantitative assessment of physical development, body composition, strength in the upper and lower limbs, speed, balance, and coordination in young children aged 2 to 3 years.

Height and weight are important indicators for the evaluation of the physical morphological development of young children. The World Health Organization has described the growth standards of height and weight in children, and their adverse changes are important risk factors for disease and even death in young children, and there also exist hysteresis effects [32]. Therefore, it is necessary to monitor the degree of development of height and weight in young children periodically. The results of this study showed that the relationship between BMI and physical health in young children was not significant, and this was confirmed by related research [22,25,33,34]. Although there is no evidence that high body weight at ages between 2 and 7 years is associated with adolescent health, a higher BMI in preschool may be associated with poor health later in life after more than a decade [35]. Therefore, monitoring of BMI is also necessary.

Gross motor skills are defined as the level of proficiency in a range of basic motor skills (e.g., kicking, walking, running), and improving gross motor skills will play an important role in growth, development, and in life, providing a foundation for the development of more specialized movements later in life [13,15]. With the development of basic motor skills entering a critical period from the age of 2 years, four physical fitness test items in this study are skill-related physical fitness. Of these, the 3 m balance walking mainly reflects the balance and coordination of children, and good walking ability is important for children’s perceptual learning, enhanced autonomy, and environmental exploration [36]. The 5 m run mainly reflects children’s ability of speed. Stair climbing requires the sensory integration of the eyes, hands, and feet and can be used to assess the hand–foot coordination in young children. The kicking a ball test is a good indication of a young child’s strength in the lower limbs. The results of this study showed that there are moderate correlations between the performance of 3 m balance walking, 5 m run and stair climbing, which may be related to the fact that all three items require the ability of muscle strength, speed, and coordination.

Handgrip strength, a measure of physical fitness that is commonly used around the world, can reflect information related to muscles, bones, or nerves. The handgrip strength of young children can also reflect their level of fine motor skills to a certain extent [37]. The results of this study showed that handgrip strength was significantly correlated with the performance of 3 m balance walking, 5 m run and stair climbing (as shown in Table 3), meaning it is possible that children’s physical performance of the upper limbs may have some relationship with their performance of the lower limbs. While there was no natural increase in the results presented in this study in terms of age, significant differences in performance were found compared to other studies of grip strength in 3–6-year-olds [22,23]. Considering that there could be differences in the comprehension ability of 2–3-year-olds, there was randomness associated with the handgrip strength test and there was a large standard deviation, which affected the results. None of the items in this set of tests were significantly correlated with gender. There were significant differences between children aged 2 and 3 years in 3 m balance walking, 5 m run, stair climbing, and kicking the ball, with 3-year-olds performing significantly better than 2-year-olds. The above characteristics of the absence of differences between genders and the presentation of age differences are consistent with the results of physical fitness tests for young children aged 3–6 years [22,23,24,26].

Retest reliability refers to the reproducibility of observed values when measurements are made repeatedly in a stable group. A high correlation was observed in both tests of handgrip strength (r = 0.859, *p* < 0.01), which showed similar reliability to studies related to 3–6-year-olds [22,37]. High reliability was demonstrated for the 3 m balance walking, 5 m run and stair climbing, and all these three test items were easily understood and accepted by young children and are also easy to perform. Although the retest reliability of kicking the ball was average, this index was still clinically meaningful, and the children also displayed some interest in this item when the test was conducted. Stair climbing and kicking the ball are examinations of young children’s overall physical ability, and kicking the ball requires a higher level of skill. The differences in test results might have been attributable to factors such as weather and field materials.

There are some limitations to this study. Firstly, compared with other age groups, comprehension, and mood changes in 2–3-year-olds may have an impact on repeat test results. Secondly, the repeatability test in this study was executed in a well-established nursery school in Shanghai, where the participants were from family backgrounds of a higher income level and the premises and management of the nursery school were of superior quality, which may affect the popularization of this physical fitness test method. Thirdly, there are certain height and width requirements on the steps used in the stair-climbing test; thus, as it is necessary to control the consistency of external conditions in the promotion of use of the method, it is difficult to standardize the equipment.

## 5. Conclusions

Overall, this study is the first to report the reliability of a field test method for physical fitness applicable to 2–3-year-olds. The reliability of this field physical test method was high in the repetitive tests of Chinese 2–3-year-olds for the four items of handgrip strength, 3 m balance walking, stair climbing and 5 m run, and the reliability was moderate for the kicking the ball item. Moreover, this test method is simple, feasible, safe, and easy to be accepted and understood by 2–3 years old children; thus, it may be used as a reference for professionals in China and abroad.

## Figures and Tables

**Figure 1 ijerph-19-07522-f001:**
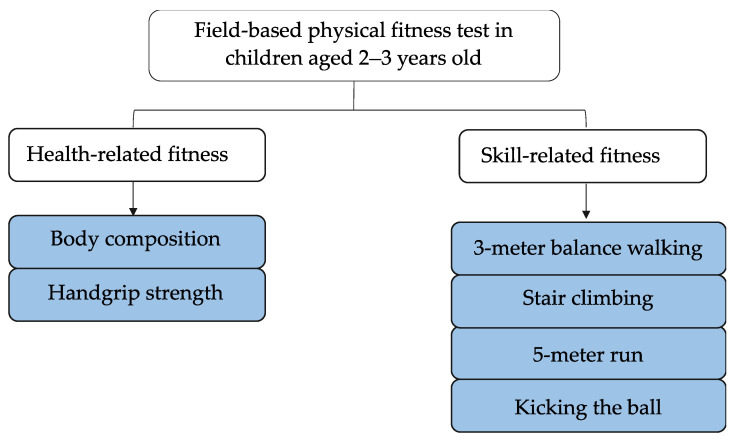
Proposal of Field-based physical fitness text for children aged 2–3 years old.

**Figure 2 ijerph-19-07522-f002:**
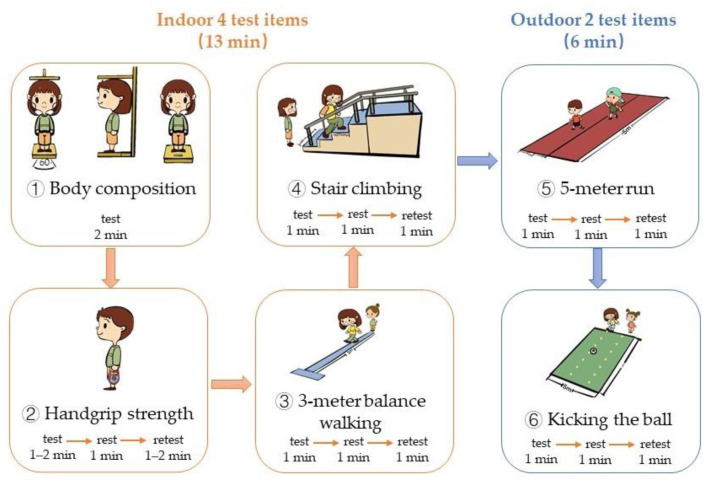
Process of Field-based physical fitness text for children aged 2–3 years old.

**Figure 3 ijerph-19-07522-f003:**
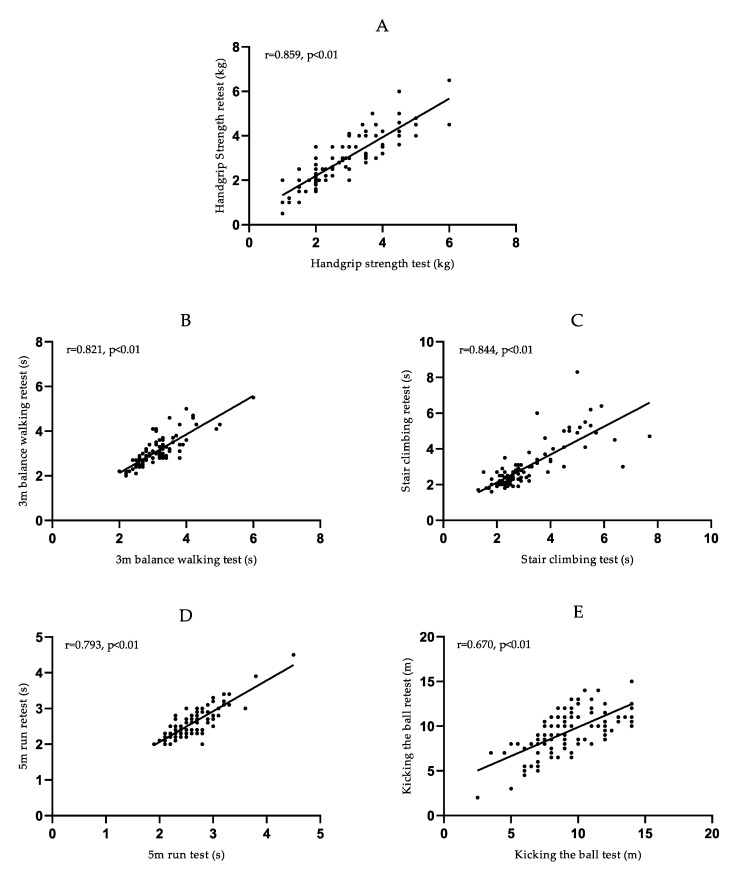
Scatterplot of the test and retest Handgrip Strength (**A**), 3 m balance walking (**B**), Stair climbing (**C**), 5 m run (**D**), Kicking the ball (**E**).

**Table 1 ijerph-19-07522-t001:** Descriptive statistics of participants considering gender and age.

	Boys (*n* = 44) Mean ± SD	Girls (*n* = 59) Mean ± SD	Sig.	2 Years (*n* = 41) Mean ± SD	3 Years (*n* = 62) Mean ± SD	Sig.	Total (*n* = 103) Mean ± SD
Age (years)	2.96 ± 0.27	3.04 ± 0.26	ns	2.74 ± 0.14	3.18 ± 0.15	**	3.01 ± 0.26
Weight (kg)	14.88 ± 1.80	14.71 ± 1.67	ns	14.21 ± 1.54	15.15 ± 1.74	**	14.78 ± 1.72
Height (cm)	96.67 ± 5.00	96.82 ± 4.15	ns	94.07 ± 3.89	98.53 ± 4.00	**	96.75 ± 4.51
Body mass index (kg/m^2^)	15.89 ± 1.02	15.66 ± 1.04	ns	16.03 ± 1.10	15.57 ± 0.95	*	15.76 ± 1.03
Handgrip strength (kg)	3.25 ± 1.30	2.94 ± 1.04	ns	2.84 ± 1.24	3.22 ± 1.09	ns	3.07 ± 1.16
3 m balance walking (s)	3.03 ± 0.66	2.93 ± 0.44	ns	3.24 ± 0.62	2.80 ± 0.41	**	2.98 ± 0.54
Stair climbing (s)	2.87 ± 1.19	2.68 ± 0.88	ns	3.20 ± 1.21	2.47 ± 0.76	**	2.76 ± 1.02
5 m run (s)	2.51 ± 0.49	2.46 ± 0.28	ns	2.60 ± 0.41	2.40 ± 0.35	**	2.48 ± 0.39
Kicking the ball (m)	10.55 ± 2.60	10.25 ± 2.49	ns	9.26 ± 2.36	11.11 ± 2.37	**	10.37 ± 2.53

Significant differences (*: *p* < 0.05, **: *p* < 0.01) between age groups; SD: standard deviation; sig.: significance; ns: no significance.

**Table 2 ijerph-19-07522-t002:** ICC of fitness test battery for children aged 2–3 years old.

	Test (*n* = 103) Mean ± SD	Retest (*n* = 103) Mean ± SD	Pearson Correlation	ICC	95% Confidence Interval
Handgrip strength (kg)	2.80 ± 1.10	2.89 ± 1.12	0.859	0.857	0.795~0.901
3 m balance walking (s)	3.13 ± 0.61	3.10 ± 0.64	0.821	0.821	0.746~0.875
Stair climbing (s)	3.05 ± 1.23	2.94 ± 1.22	0.844	0.791	0.706~0.854
5 m run (s)	2.58 ± 0.40	2.56 ± 0.41	0.793	0.844	0.779~0.892
Kicking the ball (m)	9.40 ± 2.62	9.50 ± 2.53	0.670	0.672	0.550~0.765

SD: standard deviation; ICC: intraclass correlation coefficients.

**Table 3 ijerph-19-07522-t003:** Pearson correlation between the test items.

	Weight	Height	Body Mass Index	Handgrip Strength	3 m Balance Walking	Stair Climbing	5 m Run	Kicking the Ball
Weight	1	0.804 **	0.576 **	0.190	−0.194 *	−0.041	−0.078	0.138
Height		1	−0.018	0.236 *	−0.293 **	−0.220 *	−0.217 *	0.161
Body mass index			1	0.008	0.085	0.230*	0.170	0.013
Handgrip strength				1	−0.341 **	−0.484 **	−0.404 **	0.211 *
3 m balance walking					1	0.551 **	0.576 **	−0.270 **
Stair climbing						1	0.586 **	−0.299 **
5 m run							1	−0.164
Kicking the ball								1

** Correlation is significant at the 0.01 level (2-tailed); * Correlation is significant at the 0.05 level (2-tailed).

## Data Availability

The data presented in this study are available on request from the corresponding authors.

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
