# Peer review of "Study of the Reliability of Field Test Methods for Physical Fitness in Children Aged 2–3 Years"

_ijerph, 2022, doi:10.3390/ijerph19127522_

Round 1
Reviewer 1 Report
The manuscript is interestingly written and presents the potential for evaluation of physical fitness
in children aged 2-3 years.
Introduction: The first paragraph can be minimized, the audience should know basic information about physical fitness.
The Material is well described, and The methods seem thorough. I did not find any fundamental problem that could change the results or their interpretations.
Tables and charts complement your work well.
Author Response
Point: The first paragraph can be minimized; the audience should know basic information about physical fitness.
Response: Thank you for pointing this out. We have shortened the description of physical fitness and modified it as “It comprises of two components of the physical form and physical fitness: health-related physical fitness and skill-related physical fitness”.
The revised text reads as follows on Page 1, lines 37-38.
Reviewer 2 Report
Dear authors,
First of all, thank you for this pioneering research you have done to measure the physical physical levels of 2-3 year old children. It is also interesting that you come up with meaningful and valid findings as a result of your research. Although I have some concerns about the results of your research, I hope that your answers to these concerns will be satisfactory. I will make my decision after I see the major fixes.
Introduction
Add a paragraph to the line 53-64 stating that physical fitness levels are important not only for health, but also for talent selection and monitoring of development in sports, and support this with references.
- In the last paragraph, please state your hypothesis.
Materials and Methods
-Is the number of subjects sufficient for your reliability study? Please add your Gpower results.
-First of all, you need to make a good experimental design for research. Since your subject group in the study is 2-3 years old, you should clearly explain the experimental design. For example, which tests did you do in which order and with how long rest intervals? You can also present your experimental design as a diagram.
-Create individual titles for all tests. In addition, specify your success criteria in tests separately for all tests. In particular, write down how you had the handgrip strength and 3m balance tests done to this age group, how long you waited for those who made mistakes and gave them a chance again. Because these two tests are tests that the 2-3 year old group will have difficulty in doing and cannot easily understand. In addition, if you tried to make these tests like a game, how accurate are the validity of these tests?
Discussion
You mentioned the limitations of your research in the discussion, but I would recommend you to state in the above paragraphs that the tests made in the relevant places were made for the 2-3-year-old group and that there may be errors depending on the age group in these test results.
Best
Author Response
Point 1: Add a paragraph to the line 53-64 stating that physical fitness levels are important not only for health, but also for talent selection and monitoring of development in sports, and support this with references.
Response 1: Thank you for your suggestion, and we have added the description “Being in the good physical condition is not only a prerequisite for the healthy growth of children but also a critical factor for talent selection and monitoring of development in sports.” and references 15-17.
The revised text reads as follows on P2, lines 53-54.
Point 2: In the last paragraph, please state your hypothesis.
Response 2: Thank you for your suggestion, we have modified the last paragraph as “Based on this, our study developed a set of physical fitness methods for 2-3-year-olds children according to their growth and development characteristics. This study set out to test the hypothesis that the field-based physical fitness test methods may have good reliability in 2-3 years old children, to provide a reference testing protocol for research and practice in this field”.
The revised text reads as follows on P2, lines 77-81.
Materials and Methods
Point 3: Is the number of subjects sufficient for your reliability study? Please add your G-power results.
Response 3: Thank you for pointing out this problem, and we have added the sample size calculation result is 82, and added related descriptions on P2, lines 84-85, and P6, lines 196-198.
Point 4: First of all, you need to make a good experimental design for research. Since your subject group in the study is 2-3 years old, you should clearly explain the experimental design. For example, which tests did you do in which order and with how long rest intervals? You can also present your experimental design as a diagram.
Response 4: Thank you very much for pointing out this important problem, the original version was indeed missing this content. Based on your suggestion, we have added a detailed description and a testing protocol diagram.
The revised text reads as follows on P5, lines 174-188, and Fig.2.
Point 5: Create individual titles for all tests. In addition, specify your success criteria in tests separately for all tests. In particular, write down how you had the handgrip strength and 3m balance tests done to this age group, how long you waited for those who made mistakes and gave them a chance again. Because these two tests are tests that the 2-3 year old group will have difficulty in doing and cannot easily understand. In addition, if you tried to make these tests like a game, how accurate are the validity of these tests?
Response 5: Thank you for pointing out these problems. We have revised this section and divided it into paragraphs with titles of all tests, and to describe the testing rules and considerations in detail. Please see the modifications highlighted in yellow on P3-5.
Discussion
Point 6: You mentioned the limitations of your research in the discussion, but I would recommend you to state in the above paragraphs that the tests made in the relevant places were made for the 2-3-year-old group and that there may be errors depending on the age group in these test results.
Response 6: Thanks for your important advice, and we have added this description in the limitation section: “Firstly, compared with other age groups, comprehension and mood changes in 2-3-year-olds may have a greater impact on repeat test results.”
The revised text reads as follows on P9, lines 296-298.
Reviewer 3 Report
The "Study of the reliability of field test methods for physical fitness in children aged 2-3 years" makes an important contribution to assessment of physical fitness in children in this age group.
However, some tests are debatable regarding concept of what physical fitness really is.
For example, the tests of ball kicking and balance, as well as motor coordination of arms and legs are not components of physical fitness, but are motor skills which have a correlation with physical fitness related to sports performance mainly. However, physical fitness and motor skills are fundamentally different theoretical concepts. In the case of an integration between these two concepts and healthy living habits with motor development, psychological skills, we would have something similar to the concept of motor literacy. Which in fact is not the case of the present manuscript.
I suggest that the authors separate the concepts by clearly classifying what the test battery measures according to the classic concepts of the literature. After that, the article will be ready for publication.
Author Response
Point 1: I suggest that the authors separate the concepts by clearly classifying what the test battery measures according to the classic concepts of the literature. After that, the article will be ready for publication
Response 1: We think this is an excellent suggestion, this suggestion will help us to better present the test protocol to readers. Referring to the American College of Sports Medicine's classification of physical fitness components, we have revised some relevant descriptions and added a classification diagram (Fig.1).
The revised text reads as follows on P3, lines 92-97, and Fig.1.
Round 2
Reviewer 2 Report
Thank you for effort. Your manuscript are ready to publish.